

# Genome-wide identification, evolution, and expression analysis of the *NPR1*-like gene family in pears

Yarui Wei[1], Shuliang Zhao[2], Na Liu[1] and Yuxing Zhang[1]

[1] Hebei Agricultural University, College of Horticulture, Baoding, Hebei, China
[2] Hebei University of Engineering, School of Landscape and Ecological Engineering, Handan, Hebei, China

## ABSTRACT

The NONEXPRESSOR OF PATHOGENESIS-RELATED GENES 1 (NPR1) plays a master regulatory role in the salicylic acid (SA) signal transduction pathway and plant systemic acquired resistance (SAR). Members of the *NPR1*-like gene family have been reported to the associated with biotic/abiotic stress in many plants, however the genome-wide characterization of *NPR1*-like genes has not been carried out in Chinese pear (*Pyrus bretschneideri* Reld). In this study, a systematic analysis was conducted on the characteristics of the *NPR1*-like genes in *P. bretschneideri* Reld at the whole-genome level. A total nine *NPR1*-like genes were detected which eight genes were located on six chromosomes and one gene was mapped to scaffold. Based on the phylogenetic analysis, the nine PbrNPR1-like proteins were divided into three clades (Clades I–III) had similar gene structure, domain and conserved motifs. We sorted the *cis*-acting elements into three clades, including plant growth and development, stress responses, and hormone responses in the promoter regions of *PbrNPR1*-like genes. The result of qPCR analysis showed that expression diversity of *PbrNPR1*-like genes in various tissues. All the genes were up-regulated after SA treatment in leaves except for *Pbrgene8896*. *PbrNPR1*-like genes showed circadian rhythm and significantly different expression levels after inoculation with *Alternaria alternata*. These findings provide a solid insight for understanding the functions and evolution of *PbrNPR1*-like genes in Chinese pear.

## INTRODUCTION

Pears are threatened with various diseases which were caused by fungus, bacterial, viruses, nematodes and insect bites in environment (*Robert-Seilaniantz, Grant & Jones, 2011*). These diseases limit pear quality and generate severe economic losses. Changes in phytohormones concentration or sensitivity can be triggered under biotic and abiotic stress conditions (*Pieterse et al., 2009*). Salicylic acid (SA) signaling triggers the resistance against biotrophic and hemibiotrophic pathogens (*Nie et al., 2017*), whereas a combination of Jasmonates (JA) and Ethylene (ET) signaling activates the resistance against necrotrophic pathogens (*Attaran & He, 2012*). These two pathways are mostly antagonistic: elevated biotroph resistance is often correlated with increased necrotroph

Corresponding author
Yuxing Zhang, zhyx@hebau.edu.cn

susceptibility, while elevated necrotroph resistance is correlated with enhanced susceptibility to biotrophs (*Robert-Seilaniantz, Grant & Jones, 2011*). SA has been found to be an essential in systemic acquired resistance (SAR), a broad-spectrum and long lasting plant immune response to pathogens (*An & Mou, 2011*). In SAR mediated immune response, NONEXPRESSOR OF PATHOGENESIS-RELATED GENES 1 (NPR1) is responsible for positive regulation. *PR1* gene is a molecular marker for SAR: in unchallenged cells, its transcription is repressed by TGA2 transcription factor (*Cao, Li & Dong, 1998*). In transgenic *Arabidopsis thaliana* (*A. thaliana*), overexpressing the *nahG* gene encodes salicylate hydroxylase prevented pathogen-induced accumulation of SA content and the activation of SAR. The plants showed an increase in endogenous SA, and subsequently hypersensitive response (HR) at the site of infection (*Lawton et al., 1995*).

As SA receptor, NPR1 is a transcription co-activator (*Wu et al., 2012*). The NPR1 gene encodes a protein with two protein-protein interaction domains, ankyrin repeat domain (ANK) and BTB/POZ domain (Broad Complex, Tramtrack, and Bric a Brac/Pox virus and Zinc finger). It also has a transcriptional activation domain and a nuclear localization sequence in the C terminal (*Dong, 2004*). The BTB/POZ domain is beneficial to dimerize of NPR1, and the ankyrin-repeat domain interacts with the TGA of bZIP transcription factor (*Boyle et al., 2009*; *Rochon et al., 2006*). NPR1 protein localizes both in the nucleus and cytoplasm. Before the pathgen infection, the content of SA is low in the plant, and NPR1 localizes in the cytoplasm as inactive oligomer through disulfide bonds. When a pathogen infects plant, the endogenous SA increases and the redox state changes in the cytoplasm, and NPR1 changes to active monomer, transferring from the cytoplasm to the nucleus. As transcriptional co-activator, NPR1 interacts with TGA to regulate downstream gene expression. Then PR gene expression increases and induces SAR (*Kinkema, Fan & Dong, 2000*; *Mou, Fan & Dong, 2003*).

NPR1 and its paralogues NPR3 and NPR4, were SA receptors that undertake as dominant regulators in SA-mediated local resistance and SAR (*Robert-Seilaniantz, Grant & Jones, 2011*). In transgenic apple (Fuji), overexpressing *MhNPR1* gene induced SAR and exhibited enhanced resistance to fungal diseases (*Chen et al., 2012*). Using the CRISPER/Cas9 system, *Slnpr1* mutants displayed reduced tomato drought tolerance (*Li et al., 2019*). Ectopic expression of *MuNPR1* showed enhanced scavenging ability and suppressed collapse accumulation, whereas the *MuNPR4* transgenic *Arabidopsis* were hypersensitive to *Pseudomonas syringae pv. tomato*. DC3000 (*Pst*.DC3000) infection (*Xu et al., 2019*). Recently, a new mechanism was revealed that *NPR1* facilitated its own expression by recruiting *WRKY18* and *CDK8* to the promoter of *NPR1*, leading the increased expression of the *PR* gene (*Chen et al., 2019*). *NPR1*, as SA receptor, involved in the chitosan-induced stomata closure and thus played a vital role in adjust to the adverse environments in plants (*Prodhan et al., 2020*). The proteins NPR3 and NPR4 bind SA and their function is transcriptional co-repressor and they are partly redundant in their function (*Fu et al., 2012*). *FvNPRL-1* was closer to *NPR3/NPR4* in *A. thaliana*, and ectopic expression of *FvNPRL-1* in wild type of *A. thaliana* suppressed the resistance to *Pst*.DC3000 (*Wang et al., 2018*).

Focusing on the function of redox rhythm and *NPR1* in the plant immunity and circadian clock (*Zhang et al., 2019*) proved the complex relationship between plant immune response and circadian clock. Plant circadian clocks play an important role in regulating the growth-defense balance. The nocturnal stomatal closure and active defense in the morning are good examples of how the circadian clock stop the unsustainable energy shift to immunity. Through the interaction with the circadian redox rhythm of metabolism, the circadian clock shuts off immune induction to prevent conflict with growth at night (*Dodd et al., 2005*; *Somers et al., 1998*). The circadian clock combines environmental cues with temporal information to regulate the plant physiology and development.

In *A. thaliana*, *NPR1*-like gene family has six members *NPR1*, *NPR1*-like *2* (*NPR2*), *NPR3*, *NPR4*, *BLADE-ON-PETIOLE2* (*BOP2*; *NPR5*), and *BOP1* (*NPR6*). *NPR1*, *NPR2*, *NPR3*, *NPR4* participate in the SA signal transduction pathway (*Hepworth et al., 2005*; *McKim et al., 2008*). Using bioinformatics methods, there were 17, 12, 5, and 6 *NPR1*-like genes in *Triticum aestivum*, *Triticum dicoccoides*, *Triticum urartu*, *Aegilops tauschii*, respectively (*Liu et al., 2019*). Five *NPR1*-like genes were discovered in *Persea americana* (Mill) which harbored the BTB/POZ and ankyrin repeat domains (*Backer et al., 2015*). Phylogenetic analysis divided *AtNPR1*-like gene family into three functionally distinct clades (*Peraza-Echeverria et al., 2012*; *Zhang et al., 2006*). In the first clade, *AtNPR1* and *AtNPR2* were involved in a positive regulator of SAR (*Cao et al., 1997*). In the second clade, *AtNPR3* and *AtNPR4* took part in negative SAR regulation (*Liu et al., 2004*; *Zhang et al., 2006*). In the third clade, *AtBOP1* and *AtBOP2* were involved in the organ symmetry and determinacy during the leaf morphogenesis (*Hepworth et al., 2005*; *McKim et al., 2008*). The phylogenetic analysis indicated the functional characteristics of the *NPR1*-like gene family, providing valuable resources for further study.

*NPR1*-like genes play an important role in resistance disease, growth, and development of plant tissues and organs in many plants, for examples, tomato, strawberry, avocado, banana (*Backer et al., 2015*; *Endah et al., 2008*; *Li et al., 2019*; *Wang et al., 2018*). However, *NPR1*-like gene family was still unidentified and uncharacterized in Chinese pear. In this study, it was the first report about discovering and identifying the *NPR1*-like gene family in Chinese pear genome. Also, *PbrNPR1*-like genes were analyzed the phylogeny, gene structure, conserved motif, *cis*-elements, chromosomal location, and tissue-specific expression of *P. bretschneideri* Reld. Using qPCR technology, we indicated how *NPR1*-like genes response after SA treatment and *A. alternata* infection and expression changes within 24 h. This study provided a valuable foundation for further functional analysis of the *PbrNPR1*-like genes and pear genetic improvement.

## MATERIALS & METHODS

### Plant materials and treatment

The three-year-old *P. bretschneideri* Rehd (cv.Yali) trees were cultivated in the greenhouse (16/8 h light/dark, 24 °C/20 °C day/night and relative humidity of 25%) in the experimental orchard of Hebei Agricultural University, Baoding, Hebei, China. Yali leaves were collected at 0, 1, 3, 6, 12, 24, 48, and 72 h after 0.2 mM SA treatment. Samples from

different tissue of 7-year-old Yali were gained at different stages of development. Flower buds were sampled 1 day before flowering, and then flower, young leaves, young stems, young fruit, immature leaves, old stems, mature fruits and seed were collected at 1, 15, 15, 29, 115, 115, 159 and 159 days after flowering (DAF) in 2018, respectively. The samples were taken at 8 time points throughout the day, 9:00 am, 12:00 am, 15:00 pm, 18:00 pm, 21:00 pm, 24:00 am, 3:00 am and 6:00 am. The *A. alternata* was isolated from the diseased leaves, and grew at 28 °C incubator.

## Identification of *PbrNPR1*-like gene family member

The genome of *P. bretschneideri* Rehd (cv. Dangshan Suli) was downloaded from the Genome Database for Rosaceae (GDR) (http://www.rosaceae.org/) to identify the genome-wide *NPR1*-like gene family in Chinese pear. Using hmmsearch tool in the HMMER (v3.0) software package, we detected the *PbrNPR1*-like genes in the Protein database. HMMER file according to the ankyrin domain (PF00023) and BTB/POZ domain (PF00651) was downloaded from the Pfam protein database (http://pfam.xfam.org/). The CDS length, protein size, isoelectric point (IP), molecular weight (MW) of the deduced *PbrNPR1*-like genes were analyzed using the ExPASy website (https://web.expasy.org/compute_pi/).

## Sequence alignment and phylogenetic analysis

The NPR1-like proteins sequences of rice, arabidopsis, grape, apple, pear, and other plants were obtained from NCBI and Ensamble plants (http://plants.ensembl.org/index.html) to perform alignments by using Clustal W 2.0 (*Larkin et al., 2007*). The phylogenetic tree was constructed through the neighbor-joining (NJ) method with the bootstrap value 1,000 replicates in the MEGA 6.0 software (*Tamura et al., 2013*).

## Sequence analysis of *PbrNPR1*-like gene family

The individual *PbrNPR1*-like genes structure was displayed using Gene Structure Display Server 2.0 software (GSDS, http://gsds.cbi.pku.edu.cn/) the exon/intron structure based on the alignments of *NPR1*-like genes CDS and their genomic sequences in Chinese pear and Arabidopsis (*Hu et al., 2015*). The MEME software program (http://meme-suite.org/) was used to analyze the NPR1-like protein motif with the following parameters: (1) the width of optimum motif 6-200; (2) the maximum number of motif 10 (*Bailey et al., 2009*). The domain was analyzed by using a web CD-search tool (https://www.ncbi.nlm.nih.gov/Structure/bwrpsb/bwrpsb.cgi) (*Lu et al., 2020*).

## *Cis*-elements in the promoters of *PbrNPR1*-like genes

We analyzed the promoter region up to 2,000 bp except the *Pbrgene12425* gene including 1,300 bp. The promoter sequences of the *PbrNPR1*-like genes were analyzed using PlantCARE databases (http://bioinformatics.psb.ugent.be/webtools/plantcare/html/).

## Real-time quantitative PCR (qPCR)

The total RNA of samples was extracted using the plant RNA extraction kit (Takara, Dalian, China). First-stand cDNA was synthesized through the FastQuant RT Kit with

**Table 1  Feature of *NPR1*-like genes identified in pear.**

| Gene ID | Chromosome | Start | End | Intron num | CDS (bp) | Size (aa) | MW (Da) | pI |
|---|---|---|---|---|---|---|---|---|
| Pbrgene12425 | Chr2 | 3805908 | 3808655 | 3 | 1,770 | 590 | 65,439.61 | 6.17 |
| Pbrgene33340 | Chr5 | 22753538 | 22756434 | 3 | 1,740 | 580 | 64,113.4 | 5.7 |
| Pbrgene8895 | Chr5 | 19907569 | 19910942 | 3 | 1,758 | 586 | 65,134.72 | 6.4 |
| Pbrgene8896 | Chr5 | 19914984 | 19917980 | 3 | 1,740 | 580 | 64,171.48 | 5.7 |
| Pbrgene8341 | Chr9 | 12770720 | 12774971 | 3 | 1,773 | 591 | 66,253.22 | 5.87 |
| Pbrgene6286 | Chr10 | 20497900 | 20501536 | 3 | 1,758 | 586 | 65,005.3 | 5.76 |
| Pbrgene34018 | Chr14 | 9853384 | 9856337 | 1 | 1,494 | 498 | 54,516.7 | 6.21 |
| Pbrgene2529 | Chr17 | 12796106 | 12799482 | 3 | 1,749 | 583 | 65,379.36 | 5.95 |
| Pbrgene40077 | Scaffold 008989209.1 | 75582 | 72762 | 1 | 1,494 | 498 | 54,516.7 | 6.21 |

gDNase (TianGen, Beijing, China). The qPCR was performed using the Top Green qPCR SuperMix Kit (TranGen, Beijing, China). Each 20.0 μl reaction mixture included 10.0 μl SYBR supermix, 2.0 μl cDNA template, 0.4 μl forward and reverse primer (10.0 μM) and 7.2 μl ddH$_2$O. LightCycler® 96 system (Roche, Germany) was used for qPCR using the following PCR parameters: pre-denaturation 94 °C for 30 s, followed by 42 cycles of 94 °C denaturation for 5 s, 55 °C annealing for 15 s, 72 °C extension for 10 s, melting-curve analysis was done by 95 °C for 15 s, and then a constant increase from 58 °C to 95 °C at a 2% ramp rate. All gene-specific primers for *PbrNPR1*-like gene family were designed by using DNAMAN 8.0 software except *Pbrgene33340*, and listed in Table S2 except *Pbrgene33340*. The relative expression of the *PbrNPR1*-like genes was calculated using the $2^{-\Delta\Delta CT}$ method (*Livak & Schmittgen, 2001*), and three technical replicates and three biological replicates were applied.

## Statistical analysis

The results of analysis the variance (ANOVA) and duncan multiple comparison were analyzed using SPSS software (SPSS version 17.0, Chicago, IL, USA). Data was presented in graphs through OriginPro 9.1.0 software (Microcal Software, Inc., Northampton, MA, USA).

## RESULTS

### Identification of *PbrNPR1*-like genes

After removing the redundant entries, nine PbrNPR1-like genes with typical BTB/POZ and ankryin domain were identified to unevenly distribute on the chromosomes. Among the nine *PbrNPR1*-like genes, three *PbrNPR1*-like genes were detected on chromosome 5, the other 6 genes were identified on chromosome 2, 9, 10, 14, 17, and a scaffold respectively. The characteristics of *PbrNPR1*-like genes were listed in Table 1, including the chromosomal location, intron number, coding sequence (CDS) length, amino acid length, protein molecular weight (MW) and isoelectric point (PI). The lengths of the open reading frame (ORF) of *PbrNPR1*-like genes range from 1,494 bp (*Pbrgene40077/34018*) to 1,770 bp (*Pbrgene8341*). The length of the PbrNPR1-like proteins range from 498 amino acids (Pbrgene40077/34018) to 591 amino acids (Pbrgene8341). The molecular
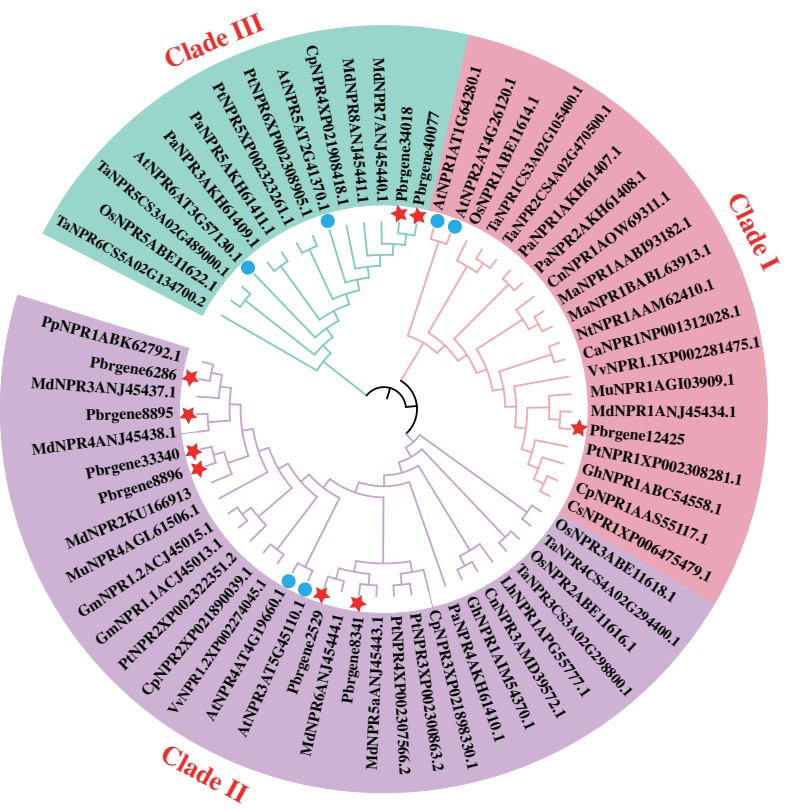

**Figure 1 Phylogenetic analysis of the NPR1-like proteins from *P. bretschneideri* Rehd and other species.** A phylogenetic tree of nine NPR1-like proteins from *P. bretschneideri* Rehd as well as other monocotyledons and dicotyledons plant species. The red star remark the NPR1-like protein in *P. bretschneider* Rehd. The blue circle remark the NPR1-like protein in *A. thaliana*. The tree was reconstructed in MEGA 6.0 software using the Neighbor-joining (NJ) tree. A thousand bootstrap replicates were performed to assess the tree reliability.

weight of these proteins range from 54.52 kDa (Pbrgene40077/34018) to 66.25 kDa (Pbrgene8341) and the PIs range from 5.7 (Pbrgene8896/33340) to 6.4 (Pbrgene8895). All genes contain one or three introns.

## Phylogenetic and protein domain analysis of P*brNPR1*-like gene family

The phylogenetic analysis of family determines the origin of the evolutionary process of the family, and the internal branch of the evolutionary tree reflects the distance of evolutionary relationships among different genes. To explore the evolutionary relationship of PbrNPR1-like gene family, NPR1-like protein sequences from 20 different monocotyledon and dicotyledon plant species were used to construct the phylogenetic tree (Table S1). According to the phylogenetic tree, these NPR1-like proteins were divided into three clades: clade I, II, III. As shown in Fig. 1, Pbrgene12425 was classified into the cluster of AtNPR1 and AtNPR2 as the positive regulator of SAR. However, the Pbrgene6286, Pbrgene8895, Pbrgene8896, Pbrgene2529, Pbrgene8341 and Pbrgene33340 involved in the clade II of AtNPR3 and AtNPR4 negative SAR regulation (*Zhang et al.,*

*2006*). The other two protein Pbrgene34018 and Pbrgene40077 were classified into the clade III along with AtBOP1 and AtBOP2 to participated in the organ determinacy and symmetry (*Hepworth et al., 2005*; *McKim et al., 2008*).

### Gene structure, motif composition, and domain of *NPR1*-like gene family

The classification of PbrNPR1-like genes was presented in Fig. 2B, in which clade I and II both contained four exons and three introns, and the Clade III (*Pbrgene34018* and *Pbrgene40077*) included two exons and one intron. The protein domain composition revealed that all NPR1-like proteins included a N-terminal BTB-POZ domain and a ankyrin repeat domain located central region (Fig. 2C). The genes in Clade I and the Clade II both have NPR1-like-C terminal region which has been proved to be an important part in NPR1-like gene family (*Boyle et al., 2009*; *Rochon et al., 2006*). The NPR1-like-C terminal protein usually existed in the eukaryotes with the size between 251 and 588 amino acids. The C-terminal region included a nuclear localization signal (NLS), a NIMINTERACTING (NIMIN) 1/2 protein binding site, and a conservative penta-amino acid motif (LENRV) (*Liu et al., 2019*). The presumed domain DUF3420 was found in eukaryotes with the length about 50 amino acids which was functionally uncharacterized. The Pbrgene2529 did not have the DUF3420 domain.

Ten conserved motifs were identified in the PbrNPR1-like gene family (Fig. S1, Table S3). As shown in Fig. 2D, Clade I and Clade II harbored motif 1–10 and Clade III featured motifs 1, 2, 3, 5, 6, 7, 9, and 10. The same conserved motif ingredients in each clade supported the polygenetic classification in the PbrNPR1-like gene family.

Multiple sequence alignments were executed to examine the conservation of residues, domains, and motifs in PbrNPR1-like and known-function AtNPR1-like (AtNPR1 to AtNPR6) (Fig. 2E). We found that npr1-2 (Cys150Tyr), nim1-2 (His300Tyr), and npr1-1 (His334Tyr) mutations in AtNPR1 were completely conserved in all PbrNPR1-like proteins. Nim1-4 (Arg432Lys) in AtNPR1 and npr4-4D (Arg419Gln) in AtNPR4 mutant sites were conserved in Clade I and Clade II. C82, C216, and C156 cysteine residues in AtNPR1 participated in its oligomer-monomer transition were highly conservation in PbrNPR1-like proteins. The Arg432 residue in AtNPR1, Arg428 in AtNPR3, and Arg419 in AtNPR4 required for their expression of SA were conserved in Clade I and Clade II in PbrNPR1-like protein.

### *Cis*-elements in the promoter regions of *PbrNPR1*-like genes

In this study, about 2,000 bp promoter region sequence was identified in the *PbrNPR1*-like gene family except for the *Pbrgene12425* with 1,300 bp. The *cis*-elements were divided into three aspects through the PlantCARE database, including plant growth and development, stress responses, and hormone responses (Table S4). In the first aspect, MRE, G-box, GTI motif, GATA motif, I-box, AE-box, ATCT-motif, TCT motif, TCCC motif and Box 4 for light responsiveness, circadian for the circadian control, $O_2$-site for zein metabolism regulation (Fig. 3) (*Abdullah et al., 2018*). In the second aspect, included a range of stress-related elements, such as, STRE and TC-rich repeats involved in stress

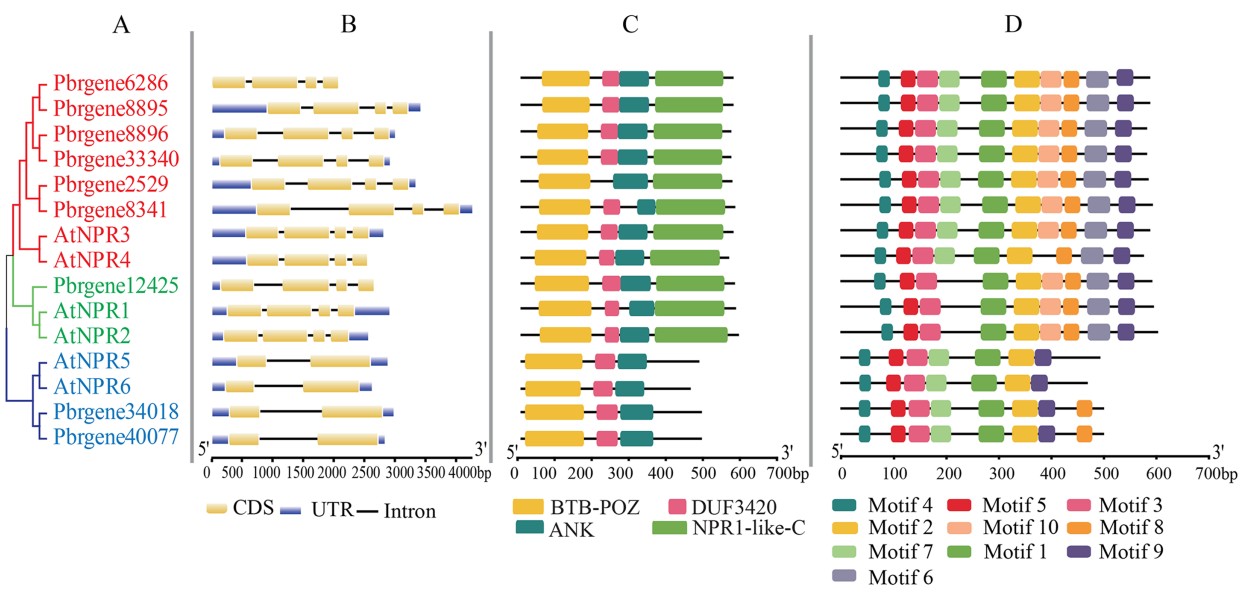

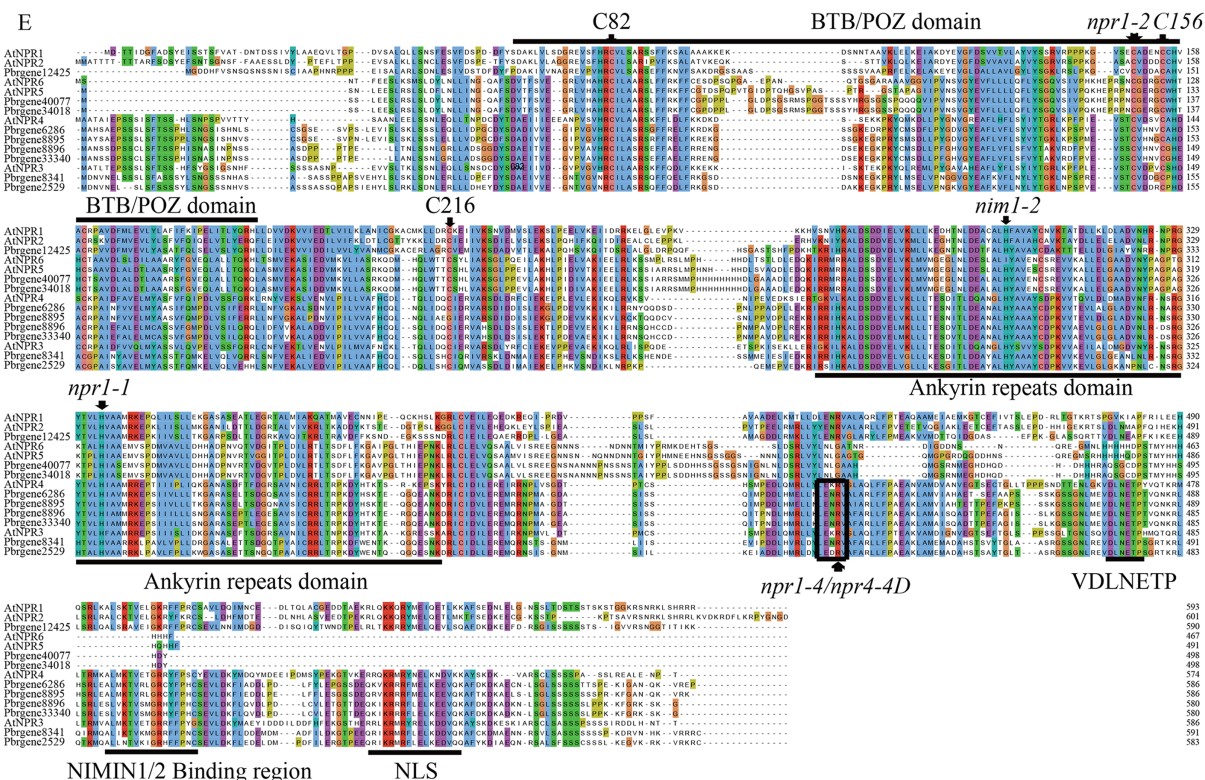

**Figure 2 Gene structure and protein sequence comparison of *PbrNPR1*-like genes with *AtNPR1*-like sequences.** (A) Phylogenetic relationship of *NPR1*-like genes in *A. thaliana* and *P. bretschneideri* Reld. (B) Exons, introns, and UTRs are showed by yellow boxes, grey lines, and blue boxes, respectively. (C) Conserved domain of the BTB/POZ, Ank, and NPR1-like C-terminal region in *A. thaliana* and *P. bretschneideri* Rehd. (D) The motifs were identified by MEME database with the protein sequences. (E) A multiple alignment of amino acid sequences of *P. bretschneideri* Rehd NPR1-like proteins (PbrNPR1 to PbrNPR6) and *A. thaliana* NPR1-like proteins (AtNPR1 to AtNPR6). npr1-1, npr1-2, nim1-2, and nim1-4 in AtNPR1 mutation sites, and highly conserved cysteines residues (C82, C216, and C156 in AtNPR1) are indicated with black arrows. The conserved domains, BTB/POZ, ANK, and some important motifs, putative hinge region (LENRV), EAR-like repression motif (VDLNETP), NIMIN-binding region, and nuclear localization signal (NLS), are indicated with solid lines.

| Gene | ACE | AE-box | ATCT-motif | AT1-motif | AT-rich element | Box 4 | Box III | G-Box | GT1-motif | HD-Zip 1 | I-box | MRE | O2-site | Sp1 | TCCC-motif | TCT-motif | chs-CMA1a | circadian | as-1 | ARE | LTR | MBS | MYB | MYC | STRE | TC-rich repeats | W box | WUN-motif | ABRE | CGTCA-motif | GARE-motif | P-box | TATC-box | TGA-element | TGACG-motif | TCA-element |
|---|---|---|---|---|---|---|---|---|---|---|---|---|---|---|---|---|---|---|---|---|---|---|---|---|---|---|---|---|---|---|---|---|---|---|---|---|
| Pbrgene12425 |  | 1 |  |  |  |  |  | 4 |  |  |  |  |  |  |  |  |  | 1 | 1 | 2 | 2 |  | 6 | 1 |  |  |  | 1 | 2 | 1 |  |  |  |  | 1 | 2 |
| Pbrgene6286 |  | 2 | 1 | 1 |  |  |  | 3 |  |  |  |  | 1 |  |  | 1 | 2 | 1 | 3 | 4 |  | 9 | 3 | 4 |  |  |  | 1 | 3 | 1 |  |  |  |  | 3 | 2 |
| Pbrgene8895 |  | 1 |  | 1 |  | 4 |  |  |  |  |  |  |  |  | 1 | 1 | 1 | 4 | 4 | 1 |  | 4 | 3 | 3 |  | 1 | 1 | 3 | 4 |  | 3 |  |  |  | 4 | 3 |
| Pbrgene8896 | 1 | 1 |  |  |  | 2 | 1 | 4 |  |  |  |  | 1 |  |  |  |  | 1 | 1 | 3 | 1 | 2 | 1 | 1 | 5 | 3 | 1 | 2 | 5 | 3 |  |  |  | 1 | 3 | 2 |
| Pbrgene2529 |  |  |  |  |  | 1 | 4 |  | 2 | 5 |  |  |  | 2 |  |  |  | 2 | 3 | 2 | 3 | 13 | 5 | 2 |  | 1 |  | 4 | 2 |  | 2 |  | 1 | 2 | 1 |  |
| Pbrgene8341 |  |  |  | 3 |  | 3 | 4 |  |  |  |  | 1 |  | 1 |  |  |  | 4 | 3 |  | 3 | 6 | 1 | 3 |  | 2 |  | 2 | 4 |  |  |  |  | 4 |  |  |
| Pbrgene34018 |  | 1 |  | 4 |  |  |  |  | 2 | 1 |  |  |  | 1 |  |  |  | 1 | 2 |  | 1 | 3 | 4 | 6 | 1 | 1 | 1 |  | 1 |  |  |  | 1 | 1 | 1 | 4 |
| Pbrgene40077 |  | 1 |  | 4 | 1 |  |  |  | 2 | 1 | 1 |  |  |  |  |  |  | 1 | 2 |  | 5 | 5 | 5 |  | 1 | 1 | 1 | 1 |  | 1 |  |  |  | 1 | 3 |  |
| Pbrgene33340 |  |  |  | 2 |  | 4 | 1 | 1 | 1 |  |  |  |  | 2 |  |  |  | 3 | 2 |  | 2 | 6 | 2 |  |  |  | 4 | 3 |  | 1 | 1 |  | 3 | 1 |  |  |

|   Plant Growth and Development   |   Stress responsive   |   Hormone Responsive   |

**Figure 3 Statistics of *cis*-acting element numbers in *NPR1*-like gene family of pear. The different numbers and colors of the grid demonstrated the numbers of different class promoter elements in these genes.** The different numbers and colors of the grid demonstrated the numbers of different class promoter elements in these genes.

responses; MBS, MYB, MYC involved drought inducibility, ARE involved in anaerobic induction, W box was a SA-induced WRKY transcription factor, The activation sequence (as-1) element took part in the transcription activation several SA-regulated PR genes (*Hernandez-Garcia & Finer, 2014*; *Yamaguchi-Shinozaki & Shinozaki, 2006*). WUN-motif related to a wound-responsive element (*Ni, Cui & Gelvin, 1996*), LTR for low-temperature responsiveness (*Xiang, Huang & Xiong, 2007*). In the third aspect, we detected the ABRE related to ABA was the most ordinary motif (*Kim et al., 2011*), CGTAC motif and TGACG motif for MeJA responsiveness, GARE-motif, P-box, and TATC box for gibberellin responsiveness, TCA-element for SA responsiveness, the TGA-element for auxin-response (*Abdullah et al., 2018*). These results indicated that *PbrNPR1*-like gene family have the advantages for enhancing abiotic stress responses and hormones responsiveness and may respond to abiotic stress and hormones.

## Expression patterns for the *PbrNPR1*-like genes in different plant tissues

To obtain a first glance at the functions of *PbrNPR1*-like genes during different developmental stages of pear, the transcript accumulation levels were observed in 10 tissues, including the leaves (April, July), stems (April, July), flowers (4 April), flowers buds (8 April), fruits (May, September), seed (September) and bud (8 March) during the budding and reproductive stages.

The result revealed that *PbrNPR1*-like genes were constitutively expressed in different tissue, and the expression level varied (Fig. 4). The expressions level of *PbrNPR1*-like genes in young fruits, mature leaves and mature stems were higher than that in mature fruits, young leaves and young stems except for *Pbrgene6268/8896/12425*. Among the eight genes, the expression level of *Pbrgene6286* gene was relatively stable in 10 tissues/ organs, *Pbrgene12425* gene was moderately expressed in leaves, *Pbrgene8895* was highly expressed in leaf, stem, seed and bud tissue, *Pbrgene8341/8896* were relatively highly

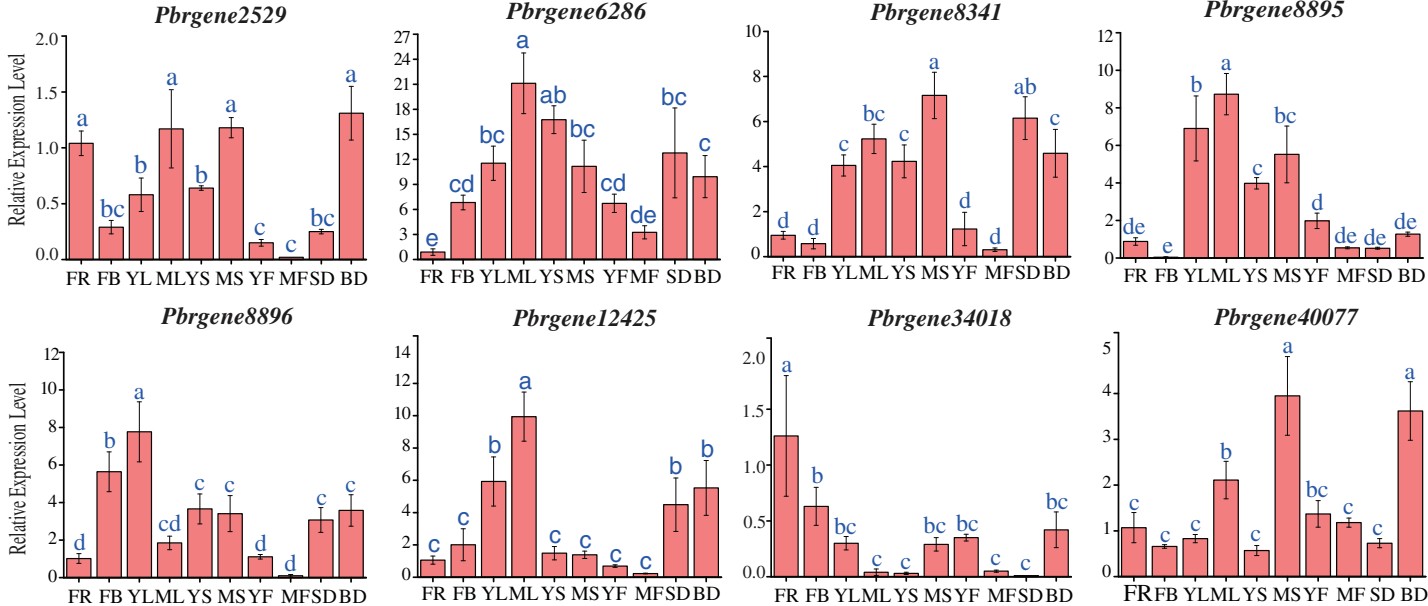

**Figure 4 Expression profile of the *PbrNPR1*-like genes in 10 different tissues.** The expression patterns of eight *PbrNPR1*-like genes in flower, flower bud, young leaf, mature leaf, young stem and mature stem, young fruit, mature fruit, seed tissues were examined by qPCR assay. FR, flower; FB, flower bud; YL, young leaf; ML, mature leaf; YS, young stem; MS, mature stem; YF, young fruit; MF, mature fruit; SD, seed; BD, bud. The error bars show the standard deviations of the three independent biological replicates. The same letter shows no significantly difference at *P* < 0.05 by Duncan's multiple range test.

expression in leaves, *Pbrgene34018/40077* were highly expressed in flower, bud and stem. The expression patterns of clade II in the 10 tissues were very similar.

## Differential expression pattern of the *PbrNPR1*-like genes in response to SA treatment

The transcript levels of *Pbrgene2529/6286/8341/8895/12425/34018/40077* were up-regulated among that the *Pbrgene2529/8341/8895/12425* reached their highest expression levels at 12 h with 2.29-fold, 2.72-fold, 6.86-fold, and 3.96-fold comparing to the 0 h samples, respectively, and then slowly returned to the baseline levels after 12 h (Fig. 5). Comparing to the 0 h samples, the expression levels of *Pbrgene6286, Pbrgene34018, Pbrgene40077* were the highest at 6 h with 4.45-fold, 3 h with 2.57-fold, 3 h with 1.97-fold, respectively. Whereas *Pbrgene8896* was first down-regulated and then up-regulated at 24 h. We observed the expression of the *Pbrgene6286/8341/12425* gene changed more than other *PbrNPR1*-like genes. In brief, the expression patterns of *PbrNPR1*-like genes were significantly affected by exogenous SA hormones.

## Expression analysis of *PbrNPR1*-like genes in response to *A. alternata*

To confirm the potential functions of the *PbrNPR1*-like gene family in response to biotic stress, the expression changes of the *PbrNPR1*-like genes were compared after challenge with *A. alternata*. All the *PbrNNPR1*-like genes can be induced by infected with *A. alternata* through varied expression patterns. The expression of *Pbrgene6286/8895/12425/34018/40077* was up-regulated and reached highest at 96 hpi or 120 hpi, meanwhile

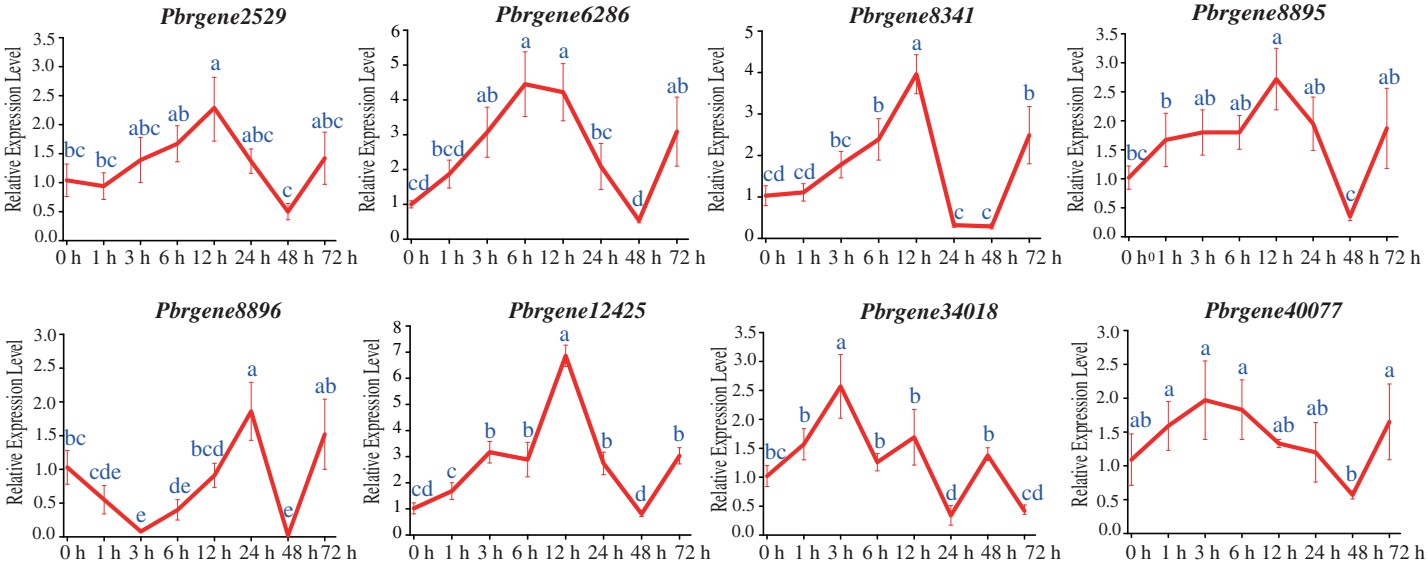

**Figure 5 Expression of *PbrNPR1*-like genes after SA treatment in Yali leaves.** The leaves were harvest at 0, 1, 3, 6, 12, 24, 48, 72 h after SA treatment. Data is means ± SD of *n* = 3 biological replicates. The same letter shows no significantly difference at *P* < 0.05 as determined by Duncan's multiple range test.

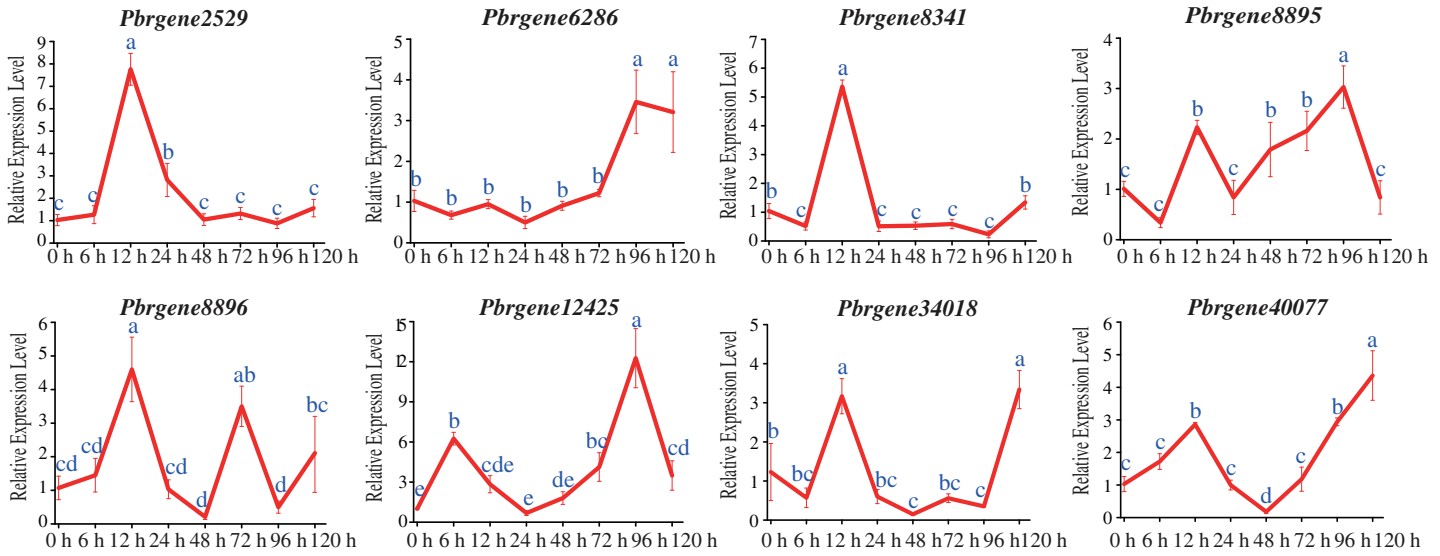

**Figure 6 Expression of *PbrNPR1*-like genes in Yali leaves after inoculation with *A. alternata*.** The leaves were harvest at 0, 6, 12, 24, 48, 72, 96, 120 h after *A. alternata* treatment. Different letters associated with each time point indicate statistically significant differences at the 5% level. The same letters indicate that the statistics did not differ significantly at *P* < 0.05 according to Duncan's multiple range tests.

these genes were reached a small peak at 12 hpi (Fig. 6). The expression of *Pbrgene12425* was significantly higher than the other *PbrNPR1*-like genes at 96 hpi. The *Pbrgene2529/8341/8896* were up-regulated and reached a peak at 12 hpi, and dramatically decreased except *Pbrgene8896* was then up-regulated at 72 hpi.

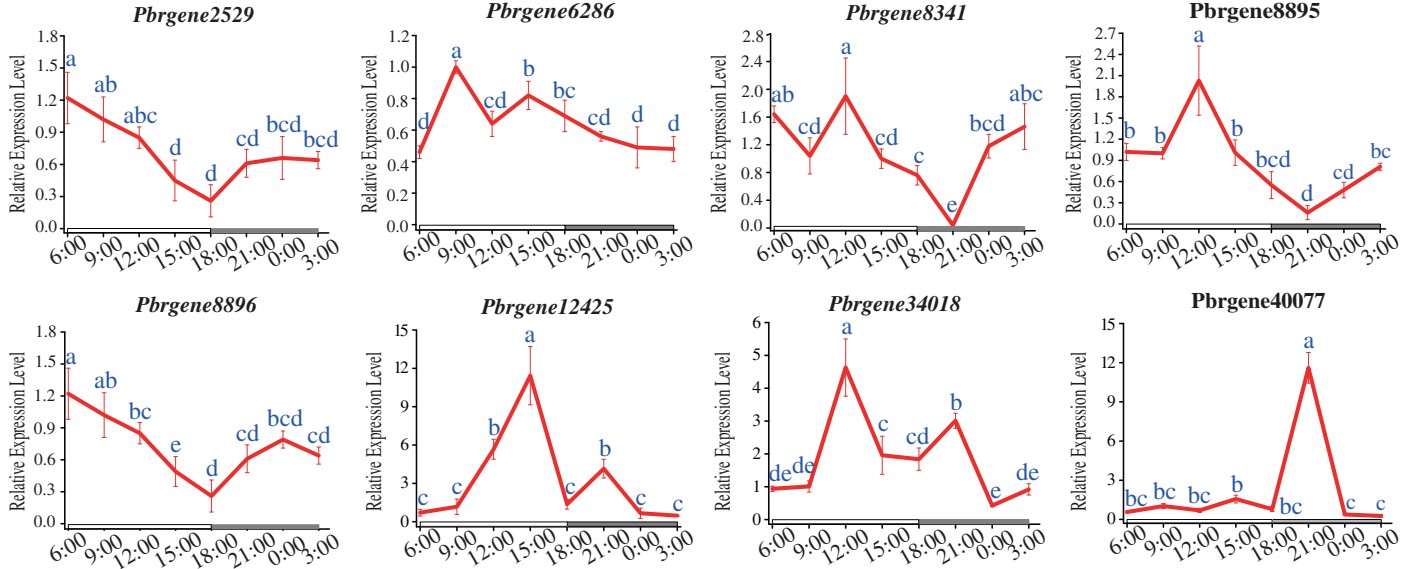

**Figure 7  Expression profile of *PbrNPR1*-like genes in Yali leaves during a day.** The leaves were collected from eight different times of a day on 25 May 2019, 9:00 am, 12:00 am, 15:00 pm, 18:00 pm, 21:00 pm, 24:00, 3:00 am and 6:00 am. Different letters associated with each time point indicate statistically significant differences at the 5% level. The same the same letters indicate that the statistics did not differ significantly at $P < 0.05$ according to Duncan's multiple range tests.

## Differential expression profile of the *PbrNPR1*-like genes in the circadian rhythm

The circadian rhythm expression patterns of *PbrNPR1*-like genes under long daytime were compared. The results showed that the expression of all the *PbrNPR1*-like genes had the phenomenon of circadian rhythm. Most of the eight genes had similar expressed patterns except for *Pbrgene40077*, which were expressed higher in the daytime than in the night (Fig. 7). The expression of *Pbrgene40077* was the highest at 21:00 pm, and low in all other time. This circadian clock phenomenon was related to the function of the family members, the clade I, II involved in plant immune, and the clade III took part in the organ symmetry and determinacy during the leaf morphogenesis.

## DISCUSSION

Nine *PbrNPR1*-like genes, *Pbrgene12425/8895/8896/8341/6286/34018/2529/33340/40077* were identified in Chinese pear. PbrNPR1-like proteins have similar gene structures, domains, and conserved motifs and amino acid residues with *NPR1*-like sequences in *Arabidopsis*, suggesting that their orthologs probably display similar biological functions in Chinese pear, while the difference between the groups specified their function was diversified. Similar results had existed in *Oryza sativa*, *Malus domestica*, *Populus trichocarpa* (*Shao et al., 2013*; *Yuan et al., 2007*; *Zhang et al., 2016*). These findings indicated that the *NPR1*-like genes were highly conserved in numerous plant species. The result of phylogenetic tree suggested that potential functional distinctions between the *PbrNPR1*-like gene family were existed.

A large number of stress response elements were existed in the promoter regions, such as ARE, LTR, MBS, MYB, MYC, STRE, TC-rich repeats, which suggested that *PbrNPR1*-like genes played an important role in the stress response process. In addition, G-box, I-box, and other optical response elements were found in the promoter regions, indicating that these genes may also be regulated by optical signals. Meanwhile, we detected the ABRE, CGTAC, TGACG, GARE, TATC box, P-box, TCA-element, and TGA-element in the nine genes, so we speculate these genes take part in the hormone-responsive and are induced by GA, SA, ABA, auxin, and MeJA hormone (*Zhao et al., 2020*). All results implied that different *PbrNPR1*-like genes played a role in special environment at different times, and thus the patterns of *PbrNPR1*-like genes responses to hormones were complex.

To further state the possible functions of the *NPR1*-like gene family in the development and growth of *P. bretschneideri* Rehd, the transcription profiles of *PbrNPR1*-like genes were studied through qPCR in 10 different tissues. The study displayed *Pbrgene12425* was expressed in all the tissues with the highest expression in mature leaves. *Pbrgene6268* and *Pbrgene8896* were expressed more higher in flower bud than that in flower. The tissue- and stage-specific expression profile implied a specific function of the two genes in the early flower development (*Shi et al., 2013*). *Pbrgene34018* and *Pbrgene40077* in clade III displayed specific expression in the leaves, stem, and flower (Fig. 4), implying that they may participated in the organ symmetry and determinacy during the leaf morphogenesis which was similar to the study on *Persea Americana*. The expression of *PaNPR1*, *PaNPR2*, and *PaNPR4* were seen in all sorts of tissues (*Backer et al., 2015*). The expression of *PaNPR3* was expressed much higher in fruit and aerial tissues. The *PaNPR5* was displayed in the roots. Therefore, database retrieval and functional prediction of *PbrNPR1*-like genes in different tissues and various stages of development indicated that *PbrNPR1*-like genes may play an important role in plant growth, and some *PbrNPR1*-like genes may have unique functions at specific stages of development.

The expression profile of the *PbrNPR1*-like genes showed obviously different after SA induction. The same results were obtained in Qinguan apple resistant disease cultivars, the *MdNPR1/2/3/4/5/6/7* were up-regulated at 6 h, and *MdNPR8* were expressed to the highest level at 12 h (*Zhang et al., 2016*). The time of reaching peak expression is different in the various plants, the possible reason is different defense system and the time of the infection and reaction varies. The expression of defense-related genes increased after 3 h in the morning with SA inducing in the morning or evening under constant light for 3 h while the expression of growth-related genes increased in the evening (*Zhou et al., 2015*). The *NPR1*-like genes would be induced after SA treatment and participated in the SA signal transduction which plays an important role in local defense and distal tissue of wide-spectrum SAR (*Baldwin & Meldau, 2013*).

Plant disease activated by the pathogens promote defense immunity to plant pathogens and the defense immunity is a difficult mechanism including the triggering of multiple immune mechanisms reaction (*Spoel & Dong, 2012*). Several resistant genes and proteins associated with the pathogenesis have been isolated and they can be used to improve the

plant defense against different diseases. The PR proteins play an important role in plant defense systems. NPR1 can interact with some members of the TGA family of bZIP transcription factors which combine with the as-1-like (TGACG) element in the PR gene promoter and take charge of PR gene expression (*Zhang et al., 2003*; *Zhou et al., 2000*). The results (Fig. 6) not only showed that there were significantly different in the expression of the family genes after infecting with *A. alternata* and some *PbrNPR1*-like genes were assciated with pear resistance to *A. alternata* in pear. *PbrNPR1*-like genes were pathogen-inducible and participated in the immune system of pear, enhancing the disease resistance.

Studies have found that the expression of *NPR1* has circadian rhythm in *A. thaliana* (*Zhou et al., 2015*), which is balanced between the regulation of self-growth and immunity. The expression of *PbrNPR1*-like gene family showed circadian rhythm in natural conditions during a day, indicating that the *PbrNPR1*-like genes may be conserved in plants circadian rhythm. The stomata on the surface of plant leaves opens in the morning, making it vulnerable to pathogen invasion, the plant activates a defense mechanism which makes it more resistant to disease in the morning than at night, the plant grows normally at night (*Korneli, Danisman & Staiger, 2014*; *Zhang et al., 2013*). This result verified the conclusion that SA mediated immune responses in the morning helped to avoid the conflicts between SA immune responses and growth-related activities that need to transport water at night (*Zhou et al., 2015*). This change in plant defense response may reflect an adaptation to change physical conditions during the day and the temperatures and humidity are generally more conducive to pathogen challenges in the morning (*Karapetyan & Dong, 2018*).

The functions of the *NPR1*-like genes maybe much more intricate in Chinese pear than in *A. thaliana*. Overexpression of *AtNPR1* can improve the disease resistance of *A. thaliana*, while *AtNPR3* and *AtNPR4* have redundant functions and opposite functions to the *AtNPR1* (*Ding et al., 2018*). When the biotic and abiotic infection, SA will combine with its receptors (NPR1, NPR3, and NPR4) and induce the expression of PR protein, which leads to trigger the immune responses. *AtNPR2* was more similar to the *AtNPR1* than the other *NPR1* orthologs and would play an important role in the SA perception and acted as an evolutive reservoir of the *NPR1* (*Castello et al., 2018*). In pear, it was speculated there were more homologs would interact with each other to regulate the balance *PbrNPR1*. The *PbrNPR1* homologs can be regulated by different transcripts or transcription factors.

This study provided the preliminary *PbrNPR1*-like gene family information and functional annotation of the nine discovered *PbrNPR1*-like genes from pear. Sequence structure, homology, and phylogenetic analysis suggested that seven *PbrNPR1*-like proteins might participate in the defense responses, the rest two genes were likely involved in tissue development. Hormone and expression in various tissues provided a support for this and allow future research to learn much more about the possible role of the *PbrNPR1*-like genes in SAR in pear. The future efforts will be focused on the localization and the intracellular interactions of defense-related *PbrNPR1*-like proteins as well as

the role of overexpressing *PbrNPR1*-like genes in the *npr1* mutant and wild-type *Arabidopsis*.

## CONCLUSIONS

Based on genomic data of *P. bretschneideri* Reld, we identified nine *PbrNPR1*-like genes. We conducted phylogenetic analyses, as well as conserved domain, conversed motif, promoter and expression profiling of the *PbrNPR1*-like gene family under SA and *A. alternata*. According to the structural and phylogenetic characteristics of the PbrNPR1-like protein sequences, they were divided into three clades. Most of the genes were responsive to SA and *A. alternata*. The expression of all the *PbrNPR1*-like genes had the phenomenon of circadian rhythm which most genes were expressed higher in the daytime than night except for the Pbrgene40077. These findings provide a solid insight for understanding the functions and evolution of *PbrNPR1*-like genes in Chinese pear. Future studies can be performed for gene function for the mechanism of resistance disease in Yali.

### Funding

This research was funded by the Technology Innovation center for Pear of Hebei Province, the Industry Technology Engineering Center for Pear, Ministry of Education, and Research foundation for introduced talents of Hebei Agricultural University (No. YJ2020057). The funders had no role in study design, data collection and analysis, decision to publish, or preparation of the manuscript.

### Grant Disclosures

The following grant information was disclosed by the authors:
Technology Innovation center.
Industry Technology Engineering Center.
Hebei Agricultural University: YJ2020057.

### Competing Interests

The authors declare that they have no competing interests.

### Author Contributions

- Yarui Wei conceived and designed the experiments, performed the experiments, analyzed the data, prepared figures and/or tables, authored or reviewed drafts of the paper, and approved the final draft.
- Shuliang Zhao conceived and designed the experiments, performed the experiments, analyzed the data, prepared figures and/or tables, and approved the final draft.
- Na Liu analyzed the data, authored or reviewed drafts of the paper, and approved the final draft.
- Yuxing Zhang conceived and designed the experiments, authored or reviewed drafts of the paper, and approved the final draft.

## Data Availability

The raw measurements are available in the Supplemental Files.

## Supplemental Information

Supplemental information for this article can be found online at http://dx.doi.org/10.7717/peerj.12617#supplemental-information.

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
