# Peer review of "Genome-wide identification, evolution, and expression analysis of the NPR1-like gene family in pears"

_PeerJ, doi:10.7717/peerj.12617_

## Round 0.1 · original submission · Major Revisions

The reviewers agree that your findings might be interesting to the community however the manuscript requires some major editing and rewriting.

Reviewer 1 ·

Basic reporting

Interesting and considerable work, but needs significant improvement. It is noted that the manuscript needs careful editing by someone with expertise in technical English editing paying particular attention to English grammar, spelling, and sentence structure so that the goals and results of the study are clear to the reader.

Experimental design

Check the below comments.

Validity of the findings

no comment

Additional comments

Manuscript entitled "Genome-wide identification, evolution, and expression analysis of the NPR1-like gene family in pears“, it has been identified nine NPR1-like genes in Chinese pear genome using bioinformatics and molecular biology approaches.This study provided evidence to assist the preliminary PbrNPR1-like gene family information and functional annotation of the nine discovered PbrNPR1-like genes from pear.Below I have highlighted some vital points.
-This potentially interesting study has been marred by an inability to communicate the finding correctly in English and should like to suggest that the authors seek the advice of someone with a good knowledge of English, preferable native speaker.
-The abstract needs to be reorganized. The abstract is wordy and needs to be simplified.
- The last paragraph of the Introduction needs to be rewritten. The results of this research can not be here.
-The results of “Pbrgene33340” were subsequently lost in Figure 4,Figure 5,Figure 6,and Figure 7.

·

Basic reporting

NPR1-like gene plays an important role in stress response. Wei et al conducted the conserved domain and phylogenetic analyses, and identified nine PbrNPR1-like genes. They analyzed the conserved domain and motif, promoter and expression profiling of the NPR1-like genes. In general, the sequence analysis of the gene family is convincing. However, I have some concerns need the author to address. The protein interactions may not be reliable and need to be conducted again. Additionally, the manuscript has many grammatical and phrasing errors, and needs to be reworded by a native speaker or professional editing service.

Experimental design

no comment

Validity of the findings

no comment

Additional comments

Abstract
The methods should be removed in the abstract. For example: ''The expression profiles of nine PbrNPR1-like genes in the different tissues were determined through qRT-PCR.''

''The PbrNPR1-like genes structure, conserved motifs, phylogenetics, chromosomal distribution, duplication events, and expression patterns were completely examined.'' These results should be described briefly in the Abstract rather than show what you did.

''The Pbrgene12425 interacted with Pbrgene6286 and Pbrgene8895'' is repeated.

There are many grammatical and phrasing errors, and the English language should be improved by a native speaker or professional editing service. For example, Line 47-51.

Line 227-231, I don’t know how the author named these PbrNPR1-like genes, I suggest the author named these genes with consecutive numbers like previous gene family research or used the gene ID directly.

Line 228-230, ''The length of the PbrNPR1-like proteins ranged from 498 amino acids (Pbrgene40077/34018) to 591 amino acids (Pbrgene12425). The molecular weight of these proteins ranged from 5.45kDa (Pbrgene40077/34018) to 6.54kDa (Pbrgene12425)'' The author had better confirm the molecular weight of these proteins again.

Line 254, ''Fig. 2D'' should be ''Fig. 2C''.

Line 254-257, ''The Clade Ⅰ (Pbrgene12425) with the Clade Ⅱ (Pbrgene6286/8895/8896/2529/8341/33340) have NPR1-like-C terminal region which was proved to be an important part in NPR1-like gene family.'' The author must cite a reference.

Line 264, ''Fig. 2C should be Fig. 2D''.

The legend of Fig. 4, Fig. 5, Fig. 6 and Fig. 7 should be more detailed.

Line 320-321, ''The expression level of Pbrgene6286 gene was the highest in this family'' How did the author know? Each of these genes was just compared to themselves.

The protein interactions make me confused, and the experiment is not reliable.
In the manuscript, the author used Pbrgene12425/6286/8895 gene names, whereas NPR1, NPR3 and NPR4 were used in the Figure 8, these must be consistent.
Line 376-377, ''Green fluorescence was observed in the cytoplasm and nucleus in these combination'' The author used the YFP fusion vector, why they observed it with Green fluorescence.
What is the marker the author used? Could the chloroplast autofluorescence (RFP) be used as the marker of cytoplasm and nucleus?
The negative control is inappropriate. I suggest the author used the YINPR1-YFPn + YFPc or YFPn + YINPR1-YFPc combination.

---

## Round 0.2 · Minor Revisions

Upon completion of the minor corrections required by reviewer 2, your article should be accepted for publication in PeerJ.

Reviewer 1 ·

Basic reporting

no comment

Experimental design

no comment

Validity of the findings

no comment

Additional comments

I suggest to accept this manuscript.

·

Basic reporting

The new version of the manuscript was greatly improved after the authors' modification. It can be accepted for publication after minor revision.
The legend of Figure 7, "the same'' was repeated.
The author should pay attention to the spaces between data and units, and spaces after punctuation throughout the manuscript, for example, line 311 21:00 pm; line 122 0 h; line 123 0.2 mM.

Experimental design

no comment

Validity of the findings

no comment

Additional comments

no comment

---

## Round 0.3 · accepted · Accept

i am please to inform you that you paper has been accepted for publication